# Phloretin Protects Bovine Rumen Epithelial Cells from LPS-Induced Injury

**DOI:** 10.3390/toxins14050337

**Published:** 2022-05-11

**Authors:** Kexin Wang, Qian Lei, Huimin Ma, Maocheng Jiang, Tianyu Yang, Qianbo Ma, Osmond Datsomor, Kang Zhan, Guoqi Zhao

**Affiliations:** Institute of Animal Culture Collection and Application, College of Animal Science and Technology, Yangzhou University, Yangzhou 225009, China; awkx5527@163.com (K.W.); lq92779277@163.com (Q.L.); mhm9401@163.com (H.M.); jmcheng1993@163.com (M.J.); tyyang0519@163.com (T.Y.); mqb1601555436@163.com (Q.M.); datsomorosmond@gmail.com (O.D.); kzhan@yzu.edu.cn (K.Z.)

**Keywords:** phloretin, lipopolysaccharide, bovine rumen epithelial cells, oxidation resistance, inflammation

## Abstract

Lipopolysaccharide (LPS) is an endotoxin that induces immune and inflammatory responses in the rumen epithelium of dairy cows. It is well-known that flavonoid phloretin (PT) exhibits anti-oxidative, anti-inflammatory and antibacterial activity. The aim of this research was to explore whether PT could decrease LPS-induced damage to bovine rumen epithelial cells (BRECs) and its molecular mechanisms of potential protective efficacy. BRECs were pretreated with PT for 2 h and then stimulated with LPS for the assessment of various response indicators. The results showed that 100 µM PT had no significant effect on the viability of 10 µg/mL LPS-induced BRECs, and this dose was used in follow-up studies. The results showed that PT pre-relieved the decline in LPS-induced antioxidant indicators (T-AOC and GSH-PX). PT pretreatment resulted in decreased interleukin-1β (IL-1β), IL-6, IL-8, tumor necrosis factor-α (TNF-α) and chemokines (CCL2, CCL5, CCL20) expression. The underlying mechanisms explored reveal that PT may contribute to inflammatory responses by regulating Toll-like receptor 4 (TLR4), nuclear transcription factor-κB p65 (NF-κB p65), and ERK1/2 (p42/44) signaling pathways. Moreover, further studies found that LPS-induced BRECs showed decreased expression of claudin-related genes (ZO-1, Occludin); these were attenuated by pretreatment with PT. These results suggest that PT enhances the antioxidant properties of BRECs during inflammation, reduces gene expression of pro-inflammatory cytokines and chemokines, and enhances barrier function. Overall, the results suggest that PT (at least in vitro) offers some protective effect against LPS-induced ruminal epithelial inflammation. Further in vivo studies should be conducted to identify strategies for the prevention and amelioration of short acute rumen acidosis (SARA) in dairy cows using PT.

## 1. Introduction

In order to maximize milk production and the economic efficiency of dairy cows, a higher proportion of concentrated feed is often added to the diet to meet nutritional needs and maintain milk production during lactation. However, high-concentrate diets fed to dairy cows contain more fermentable carbohydrates and less crude fiber, which causes ruminal acidosis with a concomitant reduction in rumen pH [1]. When dairy cows suffer from ruminal acidosis, the normal microflora is severely damaged, a large number of abnormal metabolites such as lipopolysaccharide (LPS) and biogenic amines are released, and the rumen pH value is lowered [2]. Decreased rumen pH causes the death of Gram-negative bacteria and damages the rumen epithelial barrier, triggering the translocation of endogenous LPS in the digestive tract and the release of LPS in large quantities, which then penetrates the rumen barrier into the bloodstream, thereby triggering the host’s inflammatory response [3]. The occurrence of these disease conditions not only reduces the ability of dairy cows to digest and metabolize nutrients, but also leads to a decline in milk production and milk quality, which seriously restricts the development of dairy farming [4].

LPS is a component of the cell wall of Gram-negative bacteria, which has the function of permeating the gastrointestinal barrier [5,6,7]. As a major virulence factor in *Escherichia coli*, LPS is also considered an immune stimulant because of its ability to cause endotoxin shock and induce cell apoptosis in rumen tissues [8]. LPS is a potent inducer of inflammation and can promote the expression of several inflammatory mediators, containing interleukin (IL)-1β, IL-8, IL-10 and tumor necrosis factor-alpha (TNF-α), which contributes to tissue inflammatory responses and disease development [9].

Natural compounds derived from both edible and medicinal plants continue to attract much research attention due to their multifunctional roles, including versatility and low toxicity. Phloretin (PT), a kind of flavonoid substance with the chemical composition of 3-(4-hydroxyphenyl)-1-(2,4,6-trihydroxyphenyl)-1-propanone, is one of the most widely studied compounds among natural compounds [10]. It is mainly found in the peel and root bark of apples, pears and other fruits, and exhibits antioxidant, anticancer, anti-inflammatory, and antimicrobial properties [11,12,13,14]. Of particular note, PT has been reported to have strong antibacterial activity against Gram-negative bacteria by altering the activity of key enzymes responsible for energy metabolism and redox balance within the bacteria, reducing its ability to cope with oxidative stress [15].

The rumen is the main site of nutrient digestion and absorption in ruminants, and the rumen epithelium plays a crucial physiological role in the absorption and transportation of nutrients, as well as the protection of the rumen wall [16]. The anti-inflammatory effect of PT on epithelial cells is related to NF-kB signaling [17]. PT has also been shown to improve intestinal epithelial inflammation by modulating gut microbiota [18]. It has been reported that the function of mammalian TLRs in preventing infection and controlling epithelial homeostasis depends on the recognition of microorganisms, which may explain why inflammatory cytokines and chemokines induced by TLRs lead to an adaptive immune response of pathogenic microorganisms to promote host anti-infective immunity and epithelial repair response [19]. In addition, PT induces apoptosis of human breast tumor epithelial cells (H-Ras MCF10A) in a dose-dependent manner, resulting in the inhibition of cell proliferation [20]. The anti-inflammatory and antioxidant effects of PT have been fully proved. However, few studies have examined the protective effect of PT on bovine rumen epithelial cells (BRECs) during inflammation. Therefore, understanding the protective mechanism of PT is very important for effectively maintaining rumen health and normal physiological function of dairy cows. This study was aimed at exploring whether PT could inhibit LPS-induced damage to BRECs, thereby providing a theoretical basis for animal husbandry to prevent and treat rumen inflammation. These findings may provide insights for exploring the role of PT in ruminant epithelial infection and the regulation of innate immune responses.

## 2. Results

### 2.1. Dose–Effect of Phloretin and LPS on the Viability of BRECs

As shown in Figure 1A, BRECs preincubated with different doses of PT (0, 1, 10, 20, 40, 60, 80, 100 and 1000 µM) for 12 h had no toxic effect (*p* > 0.05). Meanwhile, the apoptosis rate of BRECs treated with 100 μM PT for 12 h showed no significant effect by flow cytometry analysis (*p* > 0.05) (Figure 1B). There were no significant differences in the cell viability of BRECs stimulated by different doses of LPS (0, 1, 10, 20 and 40 μg/mL) for 3 h (*p* > 0.05). The cell viability of BRECs began to decline at varying degrees with 20 and 40 μg/mL LPS stimulation for 6 h (*p* < 0.05) (Figure 1C).

### 2.2. Effect of Phloretin on LPS-Induced Cell Viability

PT was preincubated at a working concentration of 100 μM for 2 h. As the activity of cells induced by LPS of 20 μg/mL and 40 μg/mL was significantly reduced at 6 h, we chose a challenge of LPS concentration of 10 μg/mL for 6 h as the optimal treatment conditions for further analysis. As shown in Figure 2, there was no significant difference in cell viability under LPS and PT treatments (*p* > 0.05).

### 2.3. Effect of Phloretin on LPS-Induced Oxidative Properties

To assess antioxidant activity, T-AOC, SOD, GSH-PX, and CAT were investigated to examine the function of PT. Compared with the CON group, LPS stimulation significantly reduced the activities of GSH-PX and CAT in BRECs (Figure 3B,C) (*p* < 0.05), while PT treatment significantly raised the levels of T-AOC, SOD and GSH-PX (Figure 3A–C) (*p* < 0.05). On the contrary, co-treatment of PT and LPS resulted in significantly increased concentrations of T-AOC and GSH-PX compared to the LPS-stimulated group (Figure 3A,C) (*p* < 0.05).

### 2.4. Effect of Phloretin on LPS-Induced Inflammatory Cytokine Gene Expression in BRECs

Culturing with LPS significantly increased the mRNA expression of IL-1β, IL-6, IL-8, TNF-α and TLR4 (*p* < 0.05). In contrast, compared with the LPS group, the expression of IL-1β, IL-6, IL-8, TNF-α and TLR4 was reduced in the PT + LPS-treated group (*p* < 0.05) (Figure 4).

### 2.5. Effect of Phloretin on LPS-Induced Chemokine Gene Expression in BRECs

The mRNA expression of CXCL8, CCL2, CCL5 and CCL20 was found to be stimulated by LPS (*p* < 0.05). The PT+LPS-treated group showed lower expression levels of CCL2, CCL5 and CCL20 in BRECs compared to the LPS-treated group (*p* < 0.05) (Figure 5).

### 2.6. Effect of Phloretin on LPS-Induced the Expression of p-p65 and p-p42/44 in BRECs

RT-PCR results showed that the mRNA expressions of inflammatory factors (chemokines and TLR4) were significantly increased under LPS stimulation. Therefore, the expression of phosphorylated p65 was detected by immunofluorescence and Western blotting to explore whether the NF-κB pathway was passed through. Our results indicated that the expression of phosphorylated p65 was increased in the LPS group, and the expression of p-p65 was significantly blocked by PT pretreatment (*p* < 0.05) (Figure 6). The effect of PT on the ERK1/2 signaling pathway in BRECs challenged with LPS was also investigated. As shown in Figure 6, the result indicated that p42/44 was hyperphosphorylated upon LPS stimulation. However, treatment with PT attenuated the expression of phosphorylated p42/44 (*p* < 0.05).

### 2.7. Effect of Phloretin on LPS-Induced Expression of Tight Junction Proteins in BRECs

It is speculated that PT may regulate the LPS-induced protein expression of tight junction (TJ), because PT ameliorates inflammatory response in BRECs. Therefore, the protein expression of TJ was determined. The gene levels of ZO-1 and Occludin were down-regulated in the LPS group, and preprocessing of PT significantly enhanced their mRNA expression compared with the LPS group (*p* < 0.05) (Figure 7A). The protein expression of Claudin-1, Occludin and ZO-1 was examined to confirm the role of PT in maintaining TJ. As shown in Figure 7B, the expressions of Claudin-1, Occludin and ZO-1 were reduced after LPS stimulation, while PT pretreatment partially protected the TJ of BRECs.

## 3. Discussion

A Subacute ruminal acidosis (SARA) is recognized as one of the major problems affecting animal welfare and health in intensive ruminant production systems [21]. The accumulation of microbial metabolites and bacterial cell wall fragments (such as LPS) due to rapid fermentation in the rumen is considered a potential trigger for damage to the rumen epithelial barrier and systemic inflammatory activation. LPS and histamine were found to be two main microbial metabolites in the rumen during SARA [22,23,24]. It is well known that rumen epithelial barrier function is disrupted due to disturbance of rumen environment, caused by low pH and increased concentration of LPS in rumen fluid during SARA. However, with the banning of antibiotics, alternatives and alternative therapies are urgently needed to prevent and treat rumen epithelial inflammation in dairy cows.

The occurrence and development of many diseases are linked to oxidative stress. The occurrence of oxidative stress may be a common factor leading to metabolic diseases in transitional dairy cows [25]. SOD and CAT are considered to be the main antioxidant enzymes that maintain the balance of oxidation and antioxidant in vivo [26]. Most flavonoids reduce free radicals as hydrogen donors, thereby attenuating peroxidative damage and enhancing antioxidant capacity, due to their varying content of phenolic hydroxyl groups [27]. A previous study reported that PT obviously alleviated *S. Typhimurium*-induced oxidative stress in the colon by suppressing the level of MDA and increasing the vitality of GSH-PX and SOD [28]. The effect of the present study was consistent with the result of this research. These results demonstrated that PT improved the antioxidant capacity under oxidative stress conditions, and the increased levels of T-AOC and GSH-PX demonstrated a positive impact of PT on the cellular antioxidant defense system.

Recently, a significant relationship between oxidative stress and the mRNA expression of pro-inflammatory cytokines has been reported [29]. The generally accepted fact is that oxidative stress and inflammation are pathophysiological processes associated with multiple inflammatory diseases, and that oxidative stress is defined by the excessive production of inflammatory mediators [30]. The molecular mechanisms of inflammation have been well elucidated, and a previous report suggested that activation of the TLR4 pathway is the most common event against Gram-negative bacteria such as (LPS) [31,32]. TLR4, as an upstream regulatory gene of two major pathways of NF-κB and MAPK, is an important player in the inflammatory response process. TLR4 is considered to be the primary receptor in the rumen that recognizes E. coli. The pathway by which inflammatory E. coli stimulates the immune response primarily through the binding of LPS to TLR4 and CD14, triggering the NF-κB cascade [33]. Once TLR4 is activated by LPS, NF-κB is phosphorylated and translocated from the cytosol to the nucleus, thereby activating inflammatory cytokines (IL-1β, IL-6 and TNF-α) and chemokines (CXCL8, CCL20 or CCL5) transcription [34,35,36]. Kayisoglu et al. reported that extracellular LPS was recognized or taken up by cells causing the activation of the NF-κB signaling pathway, which in turn secretes proinflammatory cytokines to induce inflammation in the digestive system [37]. Furthermore, it has been demonstrated that LPS stimulation induces high-level expression of inflammatory cytokine mRNAs in Holstein cow rumen epithelial cells and mammary epithelial cells [38,39]. Similar results were presented in our research: LPS reduced the production of proinflammatory cytokines and chemokines. Previous studies have shown that LPS-induced acute lung injury in mice was mitigated by PT through the regulation of NF-κB and MAPK pathways [40]. Huang et al. confirmed that PT protects macrophages from the inflammatory response caused by infection with virulent E. coli strains through the TLR4-Induced NF-κB pathway [15]. To confirm that pretreatment of PT alleviates inflammation in LPS-induced BRECs, the levels of inflammatory cytokines, chemokines and TLR4 mRNA in BRECs were determined. The present study results demonstrated that pretreatment with PT attenuated inflammation in LPS-induced BRECs. The inhibitory effect of PT on LPS attack may be partly explained by the downregulation of TLR4 pathway related gene expression.

Mounting evidence suggests that p65, as an important protein in the NF-κB signal transduction pathway, not only regulates the transcriptional activity, but also promotes the binding of p50 to DNA [41,42]. In general, p65 has been shown to be heterodimerized and transferred to the nucleus after NF-κB activation. Cotranscriptional activators are recruited and bind to target DNA elements, and transcription of downstream genes is activated [43]. Results from the present study demonstrated that PT inhibited the activation of NF-κB and TLR4, which is similar to a previous report of PT inhibiting the TLR4 signaling pathway, thereby alleviating inflammation in Nontypeable Haemophilus influenzae (NTHi)-infected mice [44]. Moreover, it has been reported that PT treatment significantly alleviated the DSS-induced increase in TLR4 expression and PT suppressed the NF-κB signaling, thereby the production of inflammatory cytokines and chemokines was inhibited [13,45]. Taking these findings together, these suggest that PT may inhibit the expression of the TLR4/NF-κB pathway to protect BRECs from inflammation. One possible mechanism of PT’s protective effect on rumen is to enhance epithelial immune defense by decreasing the expression of cytokines.

The intracellular signaling pathway associated with ERK was revealed to be a canonical MAPK signaling pathway in mammalian cells. ERK (p42/44) is regarded as an vital transmitter in the inflammatory signaling cascade transducing signals [46]. In this study, phosphorylation of p42/44 was activated after LPS stimulation, and interestingly, LPS-induced p-p42/44 activation was inhibited by PT. Therefore, this study speculated that PT mitigated the negative effects of LPS-induced inflammatory cytokine synthesis in BRECs by inhibiting the activation of p42/44. This finding corroborates previous studies that reported a dose-dependent inhibition of ERK1/2 phosphorylation by PT [47]. Epithelial TJ proteins are necessary to ensure a relatively stable internal environment in the rumen and reduce inflammation levels [48]. Intact epithelial barrier function is a necessary condition for maintaining homeostasis in mechanical animals. Disruption of TJ increases paracellular permeability and translocation of pathogens and harmful substances such as endotoxins, leading to tissue damage [49]. TJ is composed of more than 30 structural or functional proteins, among which ZO-1, Claudin-1 and Occludin are key proteins that preserve the physiological function of TJ [50]. Up-regulation of Claudin-1 and Claudin-4 expression has been shown to inhibit the increase in ruminal epithelial permeability caused by TJ breakage in sheep. [51]. Therefore, maintaining an appropriate expression level of TJ protein is widely considered to be an effective way to regulate rumen epithelial function. After the addition of PT, it was reported that DSS-induced significantly increased levels of ZO-1 and Occludin [45]. Furthermore, the abundance of Claudin-1, Claudin-4, and Occludin was decreased and resulted in increased rumen epithelial permeability when rumen acidosis decreased [52]. The expression of ZO-1 and Occludin was significantly decreased in LPS-induced BRECs in the present study. However, this change was ameliorated by PT therapy, suggesting that the improvement of ruminal epithelial barrier function was one of the possible mechanisms by which PT plays an anti-rumen inflammatory response role. Interestingly, it has been demonstrated that TJ function was negatively regulated by ERK1/2, and the expression of Claudin-1 and Occludin in epithelial cells were suppressed [53]. This was consistent with the results of this experiment, suggesting that PT may also regulate the expression of epithelial TJ through ERK1/2. Therefore, the protective effect exerted by PT may be due to the suppression of NF-κB-driven gene expression increases by reducing LPS-induced phosphorylation of p65, and attenuating ERK1/2 phosphorylation under LPS exposure. It was worth noting that a certain dose of PT can protect BRECs from damage by LPS, as presented by our findings, but in vivo experiments are required to determine the most appropriate dose of PT if it is to be used in livestock production.

## 4. Conclusions

Overall, the data obtained from the present study suggest the efficacy of PT in protecting BRECs from inflammation induced by LPS stimulation. Furthermore, the ability of PT to protect BRECs can be attributed to its potential to downregulate the TLR4/NF-κB pathway activated by LPS and ERK1/2 to regulate tight junction proteins. Therefore, PT may be an ideal candidate immunomodulatory and anti-inflammatory agent against diseases caused by E. coli LPS infection. This study contributes to providing further insight into the role of PT in the prevention and improvement of SARA and serves as a reference for future studies on rumen infection models in dairy cows.

## 5. Materials and Methods

### 5.1. Materials

PT (HPLC ≥ 98%) and dimethyl sulfoxide (DMSO) used as a solvent were purchased from Solarbio Science and Technology company (Solarbio, Beijing, China). *Escherichia coli* LPS serotype O55:B5 was procured from Sigma-Aldrich company (L2880, Sigma-Aldrich, St. Louis, MO, USA)**.**

### 5.2. Cell Culture

The bovine used in this study complied with the guidelines of the Institutional Animal Care and Use Committee (IACUC) of Yangzhou University. The BRECs used in this study were isolated and cultured in our laboratory [54]. Briefly, BRECs were obtained from rumen-abdominal sac tissue (depending on papillary density) of 6 to 7-month-old (206.2 ± 15.3 kg) Holstein calves in the experimental farm of Yangzhou University. Then, BRECs were immortalized, cloned and characterized. For the in vitro analyses, immortalized BRECs were grown in DMEM/F12 medium containing 10% FBS, 100 U/mL penicillin and 100 μg/mL streptomycin with 5% CO_2_ and 37 °C.

### 5.3. Treatment Methods for BRECs

PT was diluted with DMSO to various concentrations (0, 10, 20, 40, 60, 80, 100, 1000 µM). It should be noted that the final concentration of DMSO was less than 1‰ (*v*/*v*) in the treatment solution prepared above. LPS was dissolved in the medium at specific concentrations (0, 1, 10, 20, 40 µg/mL) for subsequent experiments. In addition to the concentration screening experiments, BRECs were seeded in culture plates for 24 h, pretreated with 100 µM PT for 2 h, and induced by LPS (10 μg/mL) for 6 h, then the BRECs were further assayed with each assay. The experiment consisted of four groups as follows: CON (control), LPS (lipopolysaccharide), PT (phloretin), and PT plus LPS (co-treatment).

### 5.4. Cell Viability Assay

A CCK-8 kit (Dojindo, Shanghai, China) was used to measure cell viability. Then, 5 × 10^3^ cells/well of BRECs were seeded into 96-well plates for 12 h and treated with different concentrations of PT for 12 h at 37 °C; or treated with different doses of LPS for 3 h and 6 h at 37 °C; or pretreated with PT (100 μM) for 2 h, then exposed to LPS (10 μg/mL) for 6 h for the determination of viability. After different treatments, the BRECs were washed once with phosphate-buffered solution (PBS), 10 μL of CCK-8 was added to each well, and the BRECs were then incubated at 37 °C for 3 h. An automated microplate reader (Thermo Scientific, Shanghai, China) was used to assay the absorbance at 450 nm to determine the proliferation. The cell viability was calculated as follows: (treatment group OD − blank group OD)/(CON group OD − blank group OD), where OD = optical density [55].

### 5.5. Apoptosis Assay

The levels of apoptosis were determined in BRECs with an Annexin V-FITC/PI Apoptosis Detection kit (Vazyme Biotech Co., Ltd. Nanjing, China). BRECs (1 × 10^5^) were labeled with Annexin V-FITC and PI, according to the manufacturer’s steps. Following treatment, cells were digested with trypsin without Ethylenediaminetetraacetic acid (EDTA), centrifuged to discard the supernatant, then cells were resuspended and incubated for 10 min in 5 μL Annexin V-FITC, 5 μL PI staining solution, and 400 μL of 1× Binding buffer. Fluorescence-activated cell sorting (FACS) flow cytometry (Becton Dickinson, Franklin Lakes, NJ, USA) was used to analyze the stained cells within 1 h after incubation.

### 5.6. Antioxidant Index Assay

BRECs were seeded into 6-well cells culture plates at a density of 2 × 10^5^ per well. The supernatant was collected and the contents of total antioxidant capacity (T-AOC) (HY-M0011), superoxide dismutase (SOD) (HY-M0001), glutathione peroxidase (GSH-PX) (HY-M0004), and catalase (CAT) (HY-M0018) were determined using commercial kits (Beijing Sinouk Institute of Biological Technology, Beijing, China). To measure the content of T-AOC, an appropriate amount of the standard was diluted with distilled water at different concentrations, the reagents were sequentially added to a 1.5 mL centrifuge tube and mixed well, and the absorbance was determined at 734 nm within 10 min following the manufacturer’s protocol. SOD was determined by mixing the reagents and samples thoroughly according to the kit instructions, incubating at 37 °C for 30 min, and measuring the absorbance at 560 nm. After the supernatant and the reagent were mixed, the absorbance was determined at 412 nm to obtain the concentration of GSH-PX, and the CAT activity was monitored at 405 nm.

### 5.7. RNA Extraction and Quantitative Real-Time PCR

Isolation of total RNA was performed with TRIzol reagent (Tiangen, Beijing, China). The purity of the RNA was assessed using the A260/A280 ratio. All samples measured between 1.8 and 2.0, indicating a high level of purity. Takara Biotechnology Co. Ltd. (Takara, Beijing, China) provided the reverse transcription reagents. First-strand cDNA was synthesized according to the manufacturer’s instructions. Quantitative PCR reactions were performed using the SYBR Premix Ex Taq II Kit (Takara, Beijing, China). It consists of one cycle (pre denaturation at 95 °C for 30 s), followed by forty cycles (denaturing at 95 °C for 5 s and annealing extension at 60 °C for 30 s). As shown in Table 1, the specific primers used in the quantitative PCR were listed. The reaction conditions of real-time quantitative PCR were according to previous studies in our laboratory [56]. In order to normalize the expression levels of each target gene, we used the 2^−^^ΔΔCT^ method to compare their values to the corresponding GAPDH threshold cycle (CT).

### 5.8. Immunohistochemical Analysis of NF-κB and ERK1/2

BRECs were cultured on a unique detachable chamber of a sterile Nunc Lab-Tek chamber slide system (Thermo Fisher Scientific Inc., Waltham, MA, USA) prior to staining. After adhesion, BRECs were pre-treated with 100 μM PT for 2 h, followed by incubation with 10 μg/mL LPS for 6 h. Then, BRECs were fixed with 4% paraformaldehyde for 20 min. Then, 200 μL of EDTA antigen retrieval solution was added at 95 °C for 5 min. After three PBS washes, the slides were permeabilized for 10 min at 4 °C with 0.5% Triton X-100 in PBS (Sigma-Aldrich, St. Louis, MO, USA). BRECs were blocked for 30 min in 5% horse serum at room temperature, and incubated with p-p65 (1:1600, Cell Signaling Technology Cat# 3033, RRID: AB_331284), p-ERK (1:400, Cell Signaling Technology Cat# 4370, RRID: AB_2315112) overnight at 4 °C. BRECs were then washed with PBS, incubated with Alexa Fluor 555-labeleddonkey anti-rabbit IgG (1:500, Beyotime Cat# A0453, RRID: AB_2890132) in the dark for 1 h, and stained with 200 μL DAPI for 7 min. Then, the coverslips were cleaned. After that, the slides were viewed under a fluorescence microscope (FluoView FV1200, Olympus, Tokyo, Japan).

### 5.9. Western Blotting

BRECs were seeded in 10 cm dishes (2 × 10^6^ cells/well), and the BCA kit (Beyotime Biotechnology, Shanghai, China) was used for protein concentration determination after protein collection. We separated protein samples and transferred them to PVDF membranes (PALL, Shanghai, China). Membrane sections were then incubated with 5% horse serum. The membrane sections were placed in the indicated antibodies GAPDH (1:1000, Cell Signaling Technology Cat# 5014, RRID: AB_10693448), p-p65 (1:500, Cell Signaling Technology Cat# 3033, RRID: AB_331284), p65 (1:500, Cell Signaling Technology Cat# 4764, RRID: AB_823578), ERK (1:500, Cell Signaling Technology Cat# 4695, RRID: AB_390779), p-ERK (1:1000, Cell Signaling Technology Cat# 4370, RRID: AB_2315112), Claudin-1 (1:1000, Bioss Cat# bs-10008R, RRID: AB_2915916), Occludin (1:500, Bioss Cat# bs-10011R, RRID: AB_2915915) and ZO−1 (1:2000, Proteintech Cat# 21773-1-AP, RRID: AB_10733242) at 4 °C overnight. A horseradish peroxidase-conjugated anti-rabbit immunoglobulin G (IgG)-conjugated secondary antibody (1:1000, Cell Signaling Technology Cat# 7074, RRID: AB_2099233) was used to incubate the membrane for 2 h at room temperature. Subsequently, the membrane was washed and subjected to chemiluminescence detection by Pierce Emitter Coupled Logic (ECL) Plus Western Blotting Substrate (Thermo Scientific, Waltham, MA, USA). Image J software (National Institutes of Health, Bethesda, MD, USA) was used to measure band intensities.

### 5.10. Statistical Analysis

All experiments contained three biological replicates. Experimental data between groups were analyzed using SPSS statistical software (version 19.0; SPSS Inc., Chicago, IL, USA, 2018). The apoptosis rate of cells was evaluated using an independent samples t-test. All other data were analyzed using one-way analysis of variance (ANOVA) followed by Duncan’s multiple range test. The data were expressed as mean ± standard error of the mean. Different lowercase letters (a–d) on the bar chart indicate significant differences (n = 3, *p* < 0.05).

## Figures and Tables

**Figure 1 toxins-14-00337-f001:**
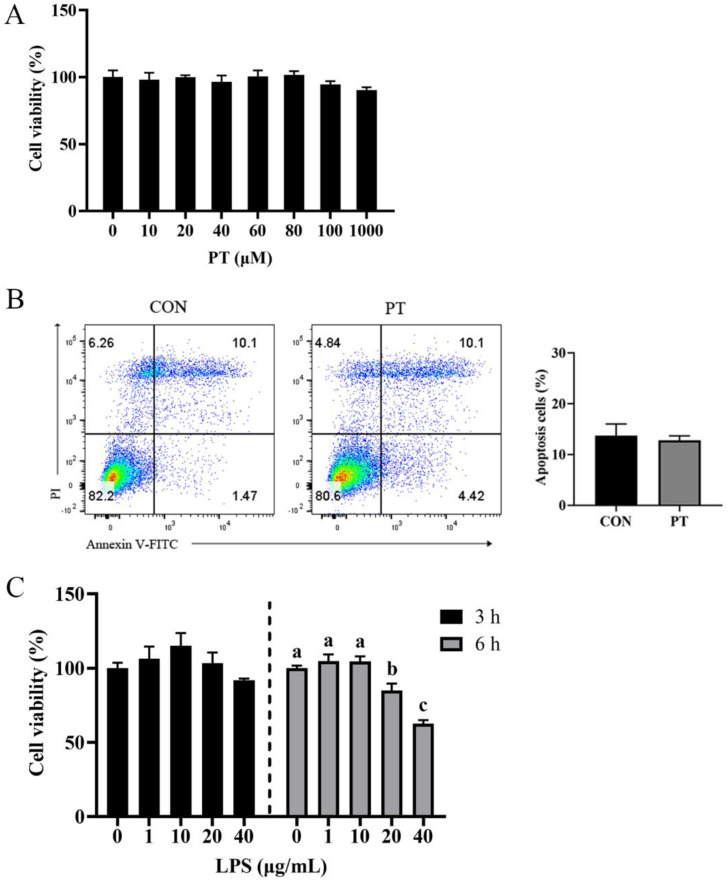
Dose–Effect of Phloretin and LPS on the viability of BRECs. (**A**) Dose–effect of different concentrations of Phloretin (0, 1, 10, 20, 40, 60, 80, 100 and 1000 µM) for 12 h on the viability of BRECs. Results are measured relative to 0 µM Phloretin (100%). (**B**) The apoptosis rate of cells treated with 100 μM PT for 12 h was measured by flow cytometry. Apoptotic cells were expressed as percentage of the total cells. (**C**) Cell viability induced by LPS at different concentrations (0, 1, 10, 20 and 40 μg/mL) and time points (3, 6 h). Results are measured relative to 0 μg/mL LPS (100%). PT, Phloretin; LPS, lipopolysaccharide. The apoptosis rate of cells was evaluated using an independent samples *t*-test. Other data were determined by ANOVA followed by Duncan’s multiple range test. All data were presented as mean ± SEM. Different lowercase letters (a–c) on the bar chart indicate significant differences (*n* = 3, *p* < 0.05).

**Figure 2 toxins-14-00337-f002:**
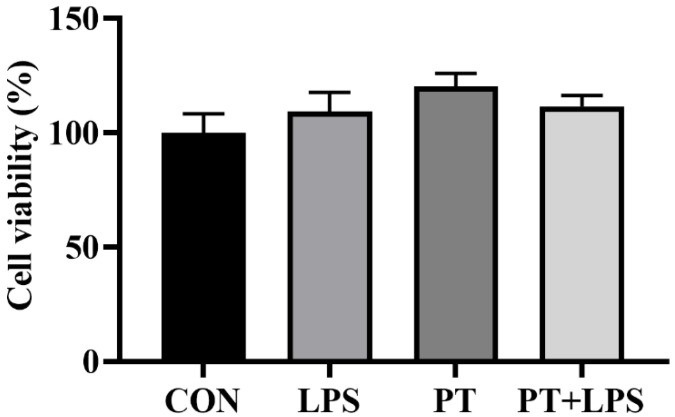
Effect of Phloretin on LPS-induced cell viability. BRECs were pretreated with Phloretin (100 μM) for 2 h and then induced by LPS (10 μg/mL) for 6 h. PT, Phloretin; LPS, lipopolysaccharide. Data were presented as a percentage of the CON group. Data as determined by ANOVA followed by Duncan’s multiple range test were presented as mean ± SEM.

**Figure 3 toxins-14-00337-f003:**
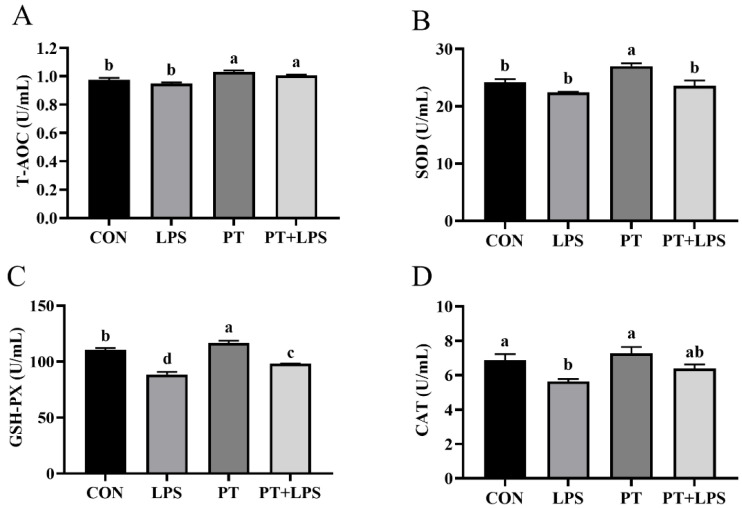
Effect of Phloretin on LPS-induced oxidative properties of BRECs. BRECs were pretreated with Phloretin (100 μM) for 2 h and then induced by LPS (10 μg/mL) for 6 h. (**A**) Total antioxidant capacity (T-AOC) activity in BRECs. (**B**) Superoxide dismutase (SOD) activity in BRECs. (**C**) Glutathione peroxidase (GSH-PX) activity in BRECs. (**D**) Catalase (CAT) activity in BRECs. PT, Phloretin; LPS, lipopolysaccharide. Data as determined by ANOVA followed by Duncan’s multiple range test were presented as mean ± SEM. Different lowercase letters (a–d) on the bar chart indicate significant differences (*n* = 3, *p* < 0.05).

**Figure 4 toxins-14-00337-f004:**
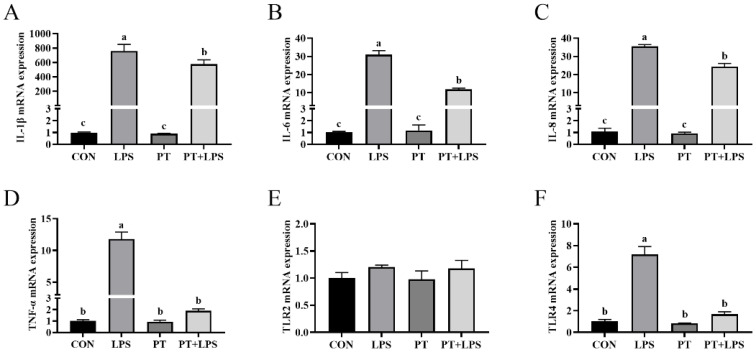
Effect of PT on LPS-induced the mRNA expression of inflammatory cytokine gene in BRECs. BRECs were pretreated with Phloretin (100 μM) for 2 h and then induced by LPS (10 μg/mL) for 6 h. (**A**) Interleukin-1β (IL-1β) mRNA level in BRECs. (**B**) IL-6 mRNA level in BRECs. (**C**) IL-8 mRNA level in BRECs. (**D**) Tumor necrosis factor-α (TNF-α) mRNA level in BRECs. (**E**) TLR2 mRNA level in BRECs. (**F**) TLR4 mRNA level in BRECs. PT, Phloretin; LPS, lipopolysaccharide. Data as determined by ANOVA followed by Duncan’s multiple range test were presented as mean ± SEM. Different lowercase letters (a–c) on the bar chart indicate significant differences (*n* = 3, *p* < 0.05).

**Figure 5 toxins-14-00337-f005:**
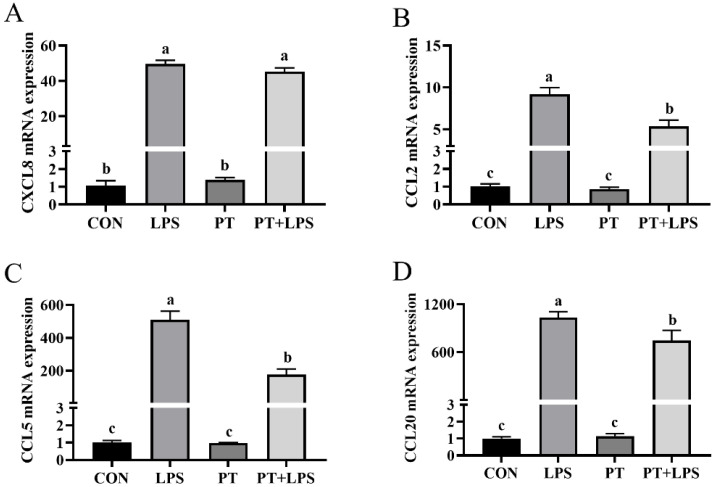
Effect of Phloretin on LPS-induced the mRNA expression of chemokine gene in BRECs. BRECs were pretreated with Phloretin (100 μM) for 2 h and then induced by LPS (10 μg/mL) for 6 h. (**A**) CXCL8 mRNA level in BRECs. (**B**) CCL2 mRNA level in BRECs. (**C**) CCL5 mRNA level in BRECs. (**D**) CCL20 mRNA level in BRECs. PT, Phloretin; LPS, lipopolysaccharide. Data as determined by ANOVA followed by Duncan’s multiple range test were presented as mean ± SEM. Different lowercase letters (a–c) on the bar chart indicate significant differences (*n* = 3, *p* < 0.05).

**Figure 6 toxins-14-00337-f006:**
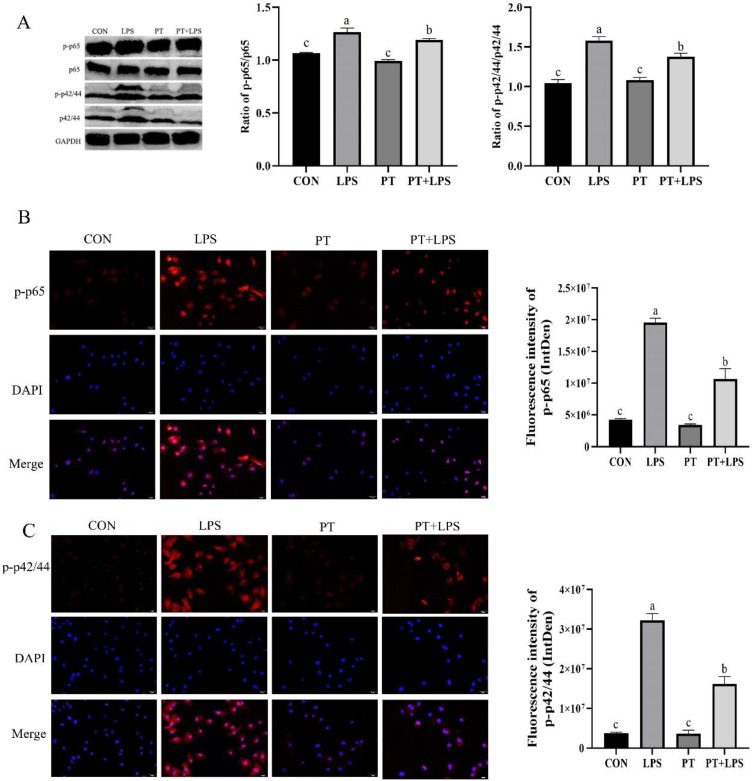
Effect of Phloretin on LPS-induced the expression of p-p65 and p-p42/44 in BRECs. BRECs were pretreated with Phloretin (100 μM) for 2 h and then induced by LPS (10 μg/mL) for 6 h. (**A**) The expression of p-p65 and p-p42/44 was determined by Western blot analysis. Immunofluorescence analysis for (**B**) p-p65 and (**C**) p-p42/44. Scale bar = 50 μm. Quantification of Fluorescence intensity of p-p65 and p-p42/44 was counted by ImageJ. PT, Phloretin; LPS, lipopolysaccharide. Data as determined by ANOVA followed by Duncan’s multiple range test were presented as mean ± SEM. Different lowercase letters (a–c) on the bar chart indicate significant differences (*n* = 3, *p* < 0.05).

**Figure 7 toxins-14-00337-f007:**
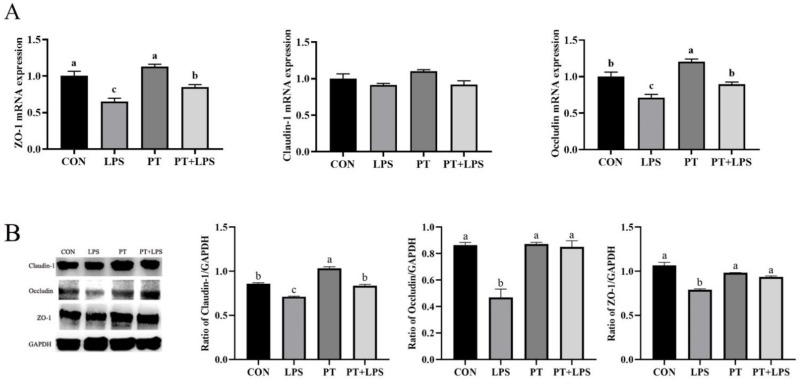
Effects of PT on LPS-induced expression of tight junction proteins in BRECs. BRECs were pretreated with Phloretin (100 μM) for 2 h and then induced by LPS (10 μg/mL) for 6 h. (**A**) ZO-1, Claudin-1 and Occludin mRNA levels in BRECs. (**B**) The expression of tight junction proteins was determined by Western blot analysis. PT, Phloretin; LPS, lipopolysaccharide. Data as determined by ANOVA followed by Duncan’s multiple range test were presented as mean ± SEM. Different lowercase letters (a–c) on the bar chart indicate significant differences (*n* = 3, *p* < 0.05).

**Table 1 toxins-14-00337-t001:** Primer sequences used for quantitative real-time PCR.

Gene	Primer Sequence	Product Size (bp)	Accession Number
CXCL8	F: TGGGCCACACTGTGAAAAT	136	NM_173925.2
	R: TCATGGATCTTGCTTCTCAGC		
CCL20	F: TTCGACTGCTGTCTCCGATA	172	NM_174263.2
	R: GCACAACTTGTTTCACCCACT		
CCL2	F: GCTCGCTCAGCCAGATGCAA	117	NM_174006.2
	R: GGACACTTGCTGCTGGTGACTC		
CCL5	F: CTGCCTTCGCTGTCCTCCTGATG	217	NM_175827.2
	R: TTCTCTGGGTTGGCGCACACCTG		
TNF-α	F: GCCCTCTGGTTCAGACACTC	192	NM_173966.3
	R: AGATGAGGTAAAGCCCGTCA		
IL-6	F: TCCTTGCTGCTTTCACACTC	129	NM_173923.2
	R: CACCCCAGGCAGACTACTTC		
IL-1β	F: CAGTGCCTACGCACATGTCT	209	NM_174093.1
	R: AGAGGAGGTGGAGAGCCTTC		
IL-8	F: TGGGCCACACTGTGAAAAT	136	NM_173925.2
	R: TCATGGATCTTGCTTCTCAGC		
TLR-4	F: GACCCTTGCGTACAGGTTGT	103	NM_174198.6
	R: GGTCCAGCATCTTGGTTGAT		
TLR-2	F: CAGGCTTCTTCTCTGTCTTGT	140	NM_174197.2
	R: CTGTTGCCGACATAGGTGATA		
ZO-1	F: TCTGCAGCAATAAAGCAGCATTTC	187	XM_010817146.1
	R: TTAGGGCACAGCATCGTATCACA		
Claudin-1	F: CGTGCCTTGATGGTGAT	102	NM_001001854.2
	R: CTGTGCCTCGTCGTCTT		
Occludin	F: GAACGAGAAGCGACTGTATC	122	NM_001082433.2
	R: CACTGCTGCTGTAATGAGG		
GAPDH	F: GGGTCATCATCTCTGCACCT	176	NM_001034034.2
	R: GGTCATAAGTCCCTCCACGA		

F: Forward; R: Reverse.

## Data Availability

Data sharing not applicable.

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
