# Peer review of "Phloretin Protects Bovine Rumen Epithelial Cells from LPS-Induced Injury"

_toxins, 2022, doi:10.3390/toxins14050337_

Round 1

Reviewer 1 Report

This article reports on the protective role of flavonoid phloretin against the detrimental effects of LPS-induced damage to bovine rumen epithelial cells of dairy cows. The authors suggest that phloretin acts via dampening the TLR4/NF-kB/ERK-signaling axis with a resulting decrease of inflammatory interleukins and chemokines as well as an enhanced expression levels of barrier constituents. The regulation of the oxidoreductase activities by LPS and the protective role of phloretin is interesting. Overall, the rationale of this study is rather convincing and it deals with a highly relevant topic. The study is structured and the manuscript is well-written. I have only a few points that the authors need to address when revising their manuscript:

  1. In the introduction, the authors should briefly discuss the influence of the gut microbiota and TLR signaling on epithelial renewal (Hörmann N, PLoS One, 2012; Rakoff-Nahoum S, Cell, 2004). This might also be part of the described eptithelial inflammatory phenotype. Is there any evidence on the impact of phloretin on the renewal of the rumen epithelium?
  2. Please indicate the number of replicates analyzed for each individual figure panel in the legends (N=?). Also specify the statistical test in the figure legend. How many times were the experiments independently repeated?
  3. In addition to the p65 translocation shown in Figure 6, the authors should detect phospho-p65 by Western blot to convincingly demonstrate activation of the NF-kB pathway.
  4. The authors should perform Western blot analyses of cell lysates to detect phloretin-dependent influences on the expected LPS-induced reduction of ZO-1, Claudin-1 and Occludin. To conclude on the barrier function, profiling of mRNA expression is not sufficient.

Author Response

  1. In the introduction, the authors should briefly discuss the influence of the gut microbiota and TLR signaling on epithelial renewal (Hörmann N, PLoS One, 2012; Rakoff-Nahoum S, Cell, 2004). This might also be part of the described eptithelial inflammatory phenotype. Is there any evidence on the impact of phloretin on the renewal of the rumen epithelium?

Thank you for your suggestion. The effects of microbial and TLRs-related signaling pathways on epithelial homeostasis have been supplemented in the Introduction based on your suggestion. It has been reported that the function of mammalian TLRs in preventing infection and controlling epithelial homeostasis relies on the recognition of microorganisms, which may explain why TLR-induced gene products such as inflammatory cytokines and chemokines are involved in host defense and Epithelial repair response. However, the effect of phloretin on rumen epithelial cells has rarely been reported, but some studies have shown that it has a certain effect on cell renewal in many endothelial cells. The specific supplementary content has been marked in the manuscript.

  1. Please indicate the number of replicates analyzed for each individual figure panel in the legends (N=?). Also specify the statistical test in the figure legend. How many times were the experiments independently repeated?

Thank you for your suggestion. I am sorry for the lack of clarity in the statistical method of the data described due to my writing mistakes. The statistical analysis section has been revised in the manuscript. Only the apoptosis data in the manuscript were analyzed by the t-test method, and the other indicators were analyzed by one-way analysis of variance (ANOVA) followed by Duncan's multiple range test. The same letters indicate no significant differences (P > 0.05), and different uppercase letters (a–d) on the top of bars indicate significant differences (n = 3, P < 0.05) vs. among the groups. Experiments were performed in triplicate, with three independent replicates in each experiment.

  1. In addition to the p65 translocation shown in Figure 6, the authors should detect phospho-p65 by Western blot to convincingly demonstrate activation of the NF-kB pathway.

Thank you for your suggestion. Western blot assays have been supplemented in the manuscript.

  1. The authors should perform Western blot analyses of cell lysates to detect phloretin-dependent influences on the expected LPS-induced reduction of ZO-1, Claudin-1 and Occludin. To conclude on the barrier function, profiling of mRNA expression is not sufficient.

Thank you for your suggestion. Due to the impact of COVID-19, the transportation of reagents and materials is difficult and experimental conditions are limited, resulting in slower experimental progress. I'm sorry for the delayed response. Immunofluorescence assays were supplemented to illustrate the expression of tight junction proteins, thereby demonstrating that PT plays a positive role in maintaining barrier function.

Reviewer 2 Report

In this work Authors tested the protective effect of phloretin (PT) against LPS toxicity in bovine rumen epithelial cells in vitro. They demonstrated that PT pre-treatment was able to recover the activity of antioxidant indicators, reducing inflammatory cytokines and chemokine overexpression induced by LPS. The involvement of TLR4, P65 NFkB and ERK1/2 was also investigated.  Additionally, the expression levels of mRNA encoding for tight junction proteins, necessary for the maintenance of epithelial barrier, were found to be recovered by PT pre-treatment of LPS-injured bovine rumen epithelial cells.

The article is well written and the contents are sound. However, in order to further improve the manuscript, some points should be addressed.

Major points

  1. Line 9-10. Authors stated that “The results showed that PT salvaged the decrease of BRECs viability induced by LPS”. However, in Fig.2 the co-treatment with LPS and PT was done only using a LPS dosage that does not affect cell viability. To demonstrate the protective effect on viability they should pre-treat cells with PT and then expose them to toxic dosage of LPS (i.e. 20ug/ml or 40ug/ml). Otherwise, this sentence should be rephrased.
  2. Statistical analyses should be better explained for each figure. It is important to add in all Figures' legends the statistical test used and to clearly explain the meaning of letters or otherwise to use asterisks. For example, Authors should indicate “a=p < 0.05 by Student’s t test vs CON; b=p < 0.05 by Student’s t test vs LPS etc”, otherwise it is difficult to understand in which way the different groups are compared.
  3. Line 85, please cite fig. 1B
  4. Fig. 1 and Fig. 2 legends, please cite the method used to quantify cell viability.
  5. Fig. 6, the magnification is too low to evaluate the subcellular localization. Please provide images at higher magnification. Additionally, for quantification images should be acquired using a confocal microscope and measuring the green fluorescence overlapped to DAPI (nuclear signal) and red fluorescence outside the nucleus (cytoplasmic signal). Fluorescence values should be expressed as Integrated Density (mean fluorescence x area measured; IntDen) corrected for the background. If quantification of confocal images cannot be performed, a subcellular fractionation isolating nuclei from cytoplasm should be performed in order to analyse by western blot the presence of NFkB p65 in the nuclear or in cytoplasmic fractions or both (see for example Fig.3 from Srivastava et al. 2018 Biomedicine & Pharmacotherapy https://doi.org/10.1016/j.biopha.2018.03.069).
  6. Fig. 7, if possible please provide a western blot assessing the increased ERK1/2 phosphorylation over the expression of total ERK1/2.
  7. Line 349, please provide codes for kits used for determining the antioxidant index and provide more information about the quantification of each one.
  8. In 5.8 section, please provide RRID number and the working dilution for each antibody.

Minor points

Please check throughout the manuscript for typos, for example:

- line 64 “within” instead of “with in”

- Fig 6.B, in the graph write “nuclei” instead of “nucles”

- Line 325, “5x 10^3” instead of “5x 103”. Similarly, in line 338 and 347.

Some phrases seems to be truncated or presenting the wrong verb, please review it:

- Line 80 “Formation in the BRECs, the effect of PT and LPS on cell proliferation was first analyzed”

- Line 231-232 “Likewise, pretreatment with PT attenuated inflammation in LPS-induced BRECs was demonstrated in this study”

- Line 241-243 “Results demonstrated that PT inhibited the activation of TLR4 and NF-κB, which was similar to the previous report that PT inhibited the TLR4 signaling pathway alleviated inflammatory in Nontypeable Haemophilus influenzae (NTHi)-infected mice”

Author Response

Due to the impact of COVID-19, the transportation of reagents and materials is difficult and experimental conditions are limited, resulting in slower experimental progress. I'm sorry for the delayed response.

  1. Line 9-10. Authors stated that “The results showed that PT salvaged the decrease of BRECs viability induced by LPS”. However, in Fig.2 the co-treatment with LPS and PT was done only using a LPS dosage that does not affect cell viability. To demonstrate the protective effect on viability they should pre-treat cells with PT and then expose them to toxic dosage of LPS (i.e. 20ug/ml or 40ug/ml). Otherwise, this sentence should be rephrased.

Thank you for your suggestion. This sentence was inappropriately written due to my mistake. It has been revised in the manuscript.

2.Statistical analyses should be better explained for each figure. It is important to add in all Figures' legends the statistical test used and to clearly explain the meaning of letters or otherwise to use asterisks. For example, Authors should indicate “a=p < 0.05 by Student’s t test vs CON; b=p < 0.05 by Student’s t test vs LPS etc”, otherwise it is difficult to understand in which way the different groups are compared.

Thank you for your suggestion. I am sorry for the lack of clarity in the statistical method of the data described due to my writing mistakes. The statistical analysis section has been revised in the manuscript. Only the apoptosis data in the manuscript were analyzed by the t-test method, and the other indicators were analyzed by one-way analysis of variance (ANOVA) followed by Duncan's multiple range test. The same letters indicate no significant differences (P > 0.05), and different uppercase letters (a–d) on the top of bars indicate significant differences (n = 3, P < 0.05) vs. among the groups.

3.Line 85, please cite fig. 1B

Thank you for your suggestion. It has been added to the manuscript.

4.Fig. 1 and Fig. 2 legends, please cite the method used to quantify cell viability.

The legend has been supplemented, and the specific calculation method for cell viability has been shown in 5.4.

5. Fig. 6, the magnification is too low to evaluate the subcellular localization. Please provide images at higher magnification. Additionally, for quantification images should be acquired using a confocal microscope and measuring the green fluorescence overlapped to DAPI (nuclear signal) and red fluorescence outside the nucleus (cytoplasmic signal). Fluorescence values should be expressed as Integrated Density (mean fluorescence x area measured; IntDen) corrected for the background. If quantification of confocal images cannot be performed, a subcellular fractionation isolating nuclei from cytoplasm should be performed in order to analyse by western blot the presence of NFkB p65 in the nuclear or in cytoplasmic fractions or both (see for example Fig.3 from Srivastava et al. 2018 Biomedicine & Pharmacotherapy https://doi.org/10.1016/j.biopha.2018.03.069).

Thank you for your suggestion. Images of larger magnifications are provided in the manuscript. In addition, immunoblotting assays for NF-KB p-p65 were supplemented in the manuscript to illustrate protein expression. I have benefited a lot from your advice. In future experiments, I will take your suggestion to take images with a confocal microscope to quantify the results.

6. Fig. 7, if possible please provide a western blot assessing the increased ERK1/2 phosphorylation over the expression of total ERK1/2.

Thank you for your suggestion. Western blot assays have been supplemented in the manuscript.

7. Line 349, please provide codes for kits used for determining the antioxidant index and provide more information about the quantification of each one.

Thank you for your suggestion. It has been added to the manuscript.

8. In Section 5.8, please provide the RRID number and working diluent for each antibody.

Thank you for your advice. It has been added to the manuscript.

Minor points

Thank you for your advice. Detailed revisions have been provided in the manuscript.

Round 2

Reviewer 1 Report

In my comment 4, I asked for Western blot quantificatons of tight junction components and not for immunoflourescence images that are difficult for quantitative analyses. Therefore, The authors should definitely perform Western blot analyses of cell lysates to detect phloretin-dependent influences on the expected LPS-induced reduction of ZO-1, Claudin-1 and Occludin. As it stands, it is impossible to make a conclusion on the barrier function. Profiling of mRNA expression is not sufficient.

Author Response

We are very grateful to your comments and thoughtful suggestions. Those comments are all valuable and very helpful for revising and improving our paper. We have adopted all of the suggestions. Western blot quantificatons of tight junction components have been performed according to your requirements.

Reviewer 2 Report

Dear Authors, thank you for improving your manuscript, I think that it is a good work. However some points should be addressed.

Major points

In Fig. 6 and in Fig. 8 immunofluorescence images should be accompanied by a proper fluorescence quantification. Additionally, in Fig.6 the fluorescence of representative image for PT+LPS does not seem to be less intense than LPS image. In possible, please provide an image that better correlate with your conclusions.

Minor points

  1. Although it is indicated in methods section, in my opinion it is better to indicate the statistical test in each figure.
  2. Additionally, please revise the figures’ legends:

- Fig.1,4,6,7 the uppercase letters in figure are a-c and not a-d.

- Fig. 2, no statistical significance is reported therefore it is unnecessary to write “different uppercase… etc.” since no letter is present in the figure.

- Throughout all figures’ legends “vs” followed by among should be omitted.

  1. Please check for typos throughout the manuscript. For example in Fig. 6 and Fig. 8, please replace “megre” with “merge”.

Author Response

We are very grateful to your comments and thoughtful suggestions. Those comments are all valuable and very helpful for revising and improving our paper. We have adopted all of the suggestions.

Round 3

Reviewer 1 Report

The authors made efforts to quantify Claudin-1, Occludin and ZO-1 protein levels by Wester blot analyses. These analyses convincingly show the LPS-induced downregulation of several tight junction proteins and the protective effect of Phloretin. Therefore, their conclusions are substantiated by the data. 

Reviewer 2 Report

Dear Authors,

thank you for improving the manuscripts.